# AI-Assisted Body Composition Assessment Using CT Imaging in Colorectal Cancer Patients: Predictive Capacity for Sarcopenia and Malnutrition Diagnosis

**DOI:** 10.3390/nu16121869

**Published:** 2024-06-14

**Authors:** Virginia Soria-Utrilla, Francisco José Sánchez-Torralvo, Fiorella Ximena Palmas-Candia, Rocío Fernández-Jiménez, Fernanda Mucarzel-Suarez-Arana, Patricia Guirado-Peláez, Gabriel Olveira, José Manuel García-Almeida, Rosa Burgos-Peláez

**Affiliations:** 1Unidad de Gestión Clínica de Endocrinología y Nutrición, Hospital Regional Universitario de Málaga, 29007 Malaga, Spain; virginiasoriau@gmail.com (V.S.-U.); fransancheztorralvo@gmail.com (F.J.S.-T.); 2Unidad de Gestión Clínica de Endocrinología y Nutrición, Hospital Universitario Virgen de la Victoria de Málaga, 29010 Malaga, Spain; rociofernandeznutricion@gmail.com (R.F.-J.); jgarciaalmeida@gmail.com (J.M.G.-A.); 3Department of Medicine and Dermatology, University of Málaga, 29016 Malaga, Spain; 4Instituto de Investigación Biomédica de Málaga (IBIMA), 29010 Malaga, Spain; 5Endocrinology and Nutrition Department, Hospital Universitari Vall d’Hebron, 08035 Barcelona, Spain; fiorellaximena.palmas@vallhebron.cat (F.X.P.-C.); fernanda.mucarzel@vhir.org (F.M.-S.-A.); rosa.burgos@vallhebron.cat (R.B.-P.); 6Diabetes and Metabolism Research Unit, Vall d’Hebron Institut De Recerca (VHIR), 08035 Barcelona, Spain; 7Department of Medicine, Universitat Autònoma de Barcelona, 08193 Barcelona, Spain; 8Department of Endocrinology and Nutrition, Quironsalud Málaga Hospital, 29004 Malaga, Spain; 9Centro de Investigación Biomédica en Red de la Fisiopatología de la Obesidad y la Nutrición (CIBERObn), Instituto de Salud Carlos III, 28029 Madrid, Spain

**Keywords:** CT imaging, colorectal cancer, malnutrition, sarcopenia, GLIM criteria, EWGSOP2 criteria, body composition, cutoff points, cancer patients

## Abstract

(1) Background: The assessment of muscle mass is crucial in the nutritional evaluation of patients with colorectal cancer (CRC), as decreased muscle mass is linked to increased complications and poorer prognosis. This study aims to evaluate the utility of AI-assisted L3 CT for assessing body composition and determining low muscle mass using both the Global Leadership Initiative on Malnutrition (GLIM) criteria for malnutrition and the European Working Group on Sarcopenia in Older People (EWGSOP2) criteria for sarcopenia in CRC patients prior to surgery. Additionally, we aim to establish cutoff points for muscle mass in men and women and propose their application in these diagnostic frameworks. (2) Methods: This retrospective observational study included CRC patients assessed by the Endocrinology and Nutrition services of the Regional University Hospitals of Malaga, Virgen de la Victoria of Malaga, and Vall d’Hebrón of Barcelona from October 2018 to July 2023. A morphofunctional assessment, including anthropometry, bioimpedance analysis (BIA), and handgrip strength, was conducted to apply the GLIM criteria for malnutrition and the EWGSOP2 criteria for sarcopenia. Body composition evaluation was performed through AI-assisted analysis of CT images at the L3 level. ROC analysis was used to determine the predictive capacity of variables derived from the CT analysis regarding the diagnosis of low muscle mass and to describe cutoff points. (3) Results: A total of 586 patients were enrolled, with a mean age of 68.4 ± 10.2 years. Using the GLIM criteria, 245 patients (41.8%) were diagnosed with malnutrition. Applying the EWGSOP2 criteria, 56 patients (9.6%) were diagnosed with sarcopenia. ROC curve analysis for the skeletal muscle index (SMI) showed a strong discriminative capacity of muscle area to detect low fat-free mass index (FFMI) (AUC = 0.82, 95% CI 0.77–0.87, *p* < 0.001). The identified SMI cutoff for diagnosing low FFMI was 32.75 cm^2^/m^2^ (Sn 77%, Sp 64.3%; AUC = 0.79, 95% CI 0.70–0.87, *p* < 0.001) in women, and 39.9 cm^2^/m^2^ (Sn 77%, Sp 72.7%; AUC = 0.85, 95% CI 0.80–0.90, *p* < 0.001) in men. Additionally, skeletal muscle area (SMA) showed good discriminative capacity for detecting low appendicular skeletal muscle mass (ASMM) (AUC = 0.71, 95% CI 0.65–0.76, *p* < 0.001). The identified SMA cutoff points for diagnosing low ASMM were 83.2 cm^2^ (Sn 76.7%, Sp 55.3%; AUC = 0.77, 95% CI 0.69–0.84, *p* < 0.001) in women and 112.6 cm^2^ (Sn 82.3%, Sp 58.6%; AUC = 0.79, 95% CI 0.74–0.85, *p* < 0.001) in men. (4) Conclusions: AI-assisted body composition assessment using CT is a valuable tool in the morphofunctional evaluation of patients with colorectal cancer prior to surgery. CT provides quantitative data on muscle mass for the application of the GLIM criteria for malnutrition and the EWGSOP2 criteria for sarcopenia, with specific cutoff points established for diagnostic use.

## 1. Introduction

Colorectal cancer (CRC) is the third most common malignant neoplasm globally and the second leading cause of cancer-related mortality [1]. This neoplasm and its surgical treatment induce a systemic inflammatory state that triggers a metabolic stress response with hypercatabolism, contributing to malnutrition [2,3]. Unfortunately, up to 60% of CRC patients suffer from disease-related malnutrition [4]. Malnutrition is an independent risk factor for increased complications and mortality [5], prolonged hospital stays, and higher postsurgery costs. Moreover, malnutrition can accelerate disease progression and functional decline [6]. This process is also closely related to a reduction in muscle mass and function, leading to sarcopenia. Previous studies have linked functional deficits and sarcopenia with a higher prevalence of postoperative complications and increased mortality [7,8,9].

Nonetheless, there is no singular parameter that precisely captures both muscle mass and nutritional status. Therefore, nutritional assessment parameters and body composition techniques that are practical, sensitive, and specific, while aligning with other malnutrition diagnostic tools, are of significant importance [10]. Among these techniques, computed tomography (CT) is considered a highly precise tool for evaluating body composition. Although its routine use for this purpose is not feasible due to the radiation dose involved, images acquired for other medical purposes can be utilized to assess body composition [11]. In fact, the use of CT for evaluating body composition has become an appropriate approach in oncology, as many patients undergo this scan for diagnostic or monitoring purposes. Some studies have validated the use of CT for evaluating body composition in cancer patients, using the level of the third lumbar vertebra (L3) as a reference point [12]. Indeed, it has already been used in patients with colorectal cancer [13,14,15]. However, to our knowledge, there are no studies evaluating the predictive capacity of body composition by CT to determine low muscle mass applying both the Global Leadership Initiative on Malnutrition (GLIM) criteria for malnutrition and the European Working Group on Sarcopenia in Older People (EWGSOP2) criteria for sarcopenia. Improved morphofunctional assessment tools could enhance the evaluation of body composition and nutritional status, thereby potentially improving clinical outcomes for CRC patients.

Our hypothesis is that body composition assessed by CT at the L3 level in CRC patients could be a valuable technique for determining low muscle mass applying both GLIM criteria for malnutrition and EWGSOP2 criteria for sarcopenia. Therefore, the aim of our study is to evaluate the utility of AI-assisted L3 CT for assessing body composition and determining low muscle mass using both GLIM criteria for malnutrition and EWGSOP2 criteria for sarcopenia in CRC patients before surgery. Additionally, we aim to describe cutoff points for muscle mass in men and women and propose their utilization in the application of this diagnostic consensus.

## 2. Materials and Methods

This retrospective observational study included CRC patients assessed by the Endocrinology and Nutrition services of the Regional University Hospitals of Malaga, Virgen de la Victoria of Malaga, and Vall d’Hebrón of Barcelona, between October 2018 and July 2023. Inclusion criteria encompassed outpatient individuals with a diagnosis of colorectal neoplasia who were attended to at the Nutrition Unit before colorectal surgery. Exclusion criteria excluded subjects who had undergone their last CT scan more than three months prior to the consultation or patients hospitalized at the time of assessment.

### 2.1. Screening, Assessment, and Nutritional Intervention

Various data were collected from these patients during a visit to the Nutrition Consultation before colorectal surgery. Oncological variables were documented, including the type of neoplasia and disease stage.

Nutritional assessment included the following:Height, weight, and body mass index (BMI), as well as weight loss in the previous six months.Morphofunctional assessment:
-Impedance measurement using a portable device (Akern BIA-101/Nutrilab analyzer, Akern SRL, Pontassieve, Florence, Italy). The variables collected were fat mass, fat-free mass, fat-free mass index (FFMI), appendicular skeletal muscle mass (ASMM), phase angle, and body cell mass (BCM).-Handgrip strength measurement using a Jamar dynamometer (Asimow Engineering Co., Los Angeles, CA, USA). Low handgrip strength was considered when values were below the cut-off points indicated in the EWGSOP2 criteria for sarcopenia [16].Diagnosis of malnutrition using GLIM criteria. It was estimated that all patients presented at least one etiological criterion, as they had colorectal neoplasia (considered a chronic inflammatory process). For the phenotypic criteria, patients with low BMI, weight loss greater than 5% in six months, and low FFMI by BIA were identified according to the cutoff points recommended by the consensus itself [3].Diagnosis of sarcopenia according to EWGSOP2 criteria. Cutoff points for low handgrip strength stipulated by the consensus were used, and ASMM assessed by BIA was used as a parameter for low muscle mass [16].

### 2.2. Body Composition Assessment by CT

To determine skeletal muscle and abdominal adipose tissue area, transverse CT images with a cut at the third lumbar vertebra (L3) were analyzed using FocusedON-BC software (version 1.0). This software features a user-friendly interface and incorporates an AI-assisted semiautomatic labeling tool, enabling users to adjust the body mass segmentation automatically performed by the software (Figure 1).

Previously conducted diagnostic CT scans of the patients were utilized, provided they were conducted no more than 3 months prior to the assessment in the Nutrition Consultation. Muscles included in the analysis comprised the psoas, erector spinae, quadratus lumborum, transversus abdominis, external and internal obliques, and rectus abdominis. Adipose tissue was also evaluated and categorized into subcutaneous, visceral, and intramuscular.

The following variables were recorded: skeletal muscle area or SMA (cm^2^ and %), skeletal muscle index or SMI (cm^2^/m^2^), intramuscular adipose tissue area or IMAT (cm^2^ and %), intramuscular adipose tissue index or IIMAT (cm^2^/m^2^), visceral fat area (VFA) (cm^2^ and %), subcutaneous fat area (SFA) (cm^2^ and %), visceral fat index (VFI) (cm^2^/m^2^), subcutaneous fat index (SFI) (cm^2^/m^2^), and mean Hounsfield units (HU) value for each segmented tissue.

For the estimation of total muscle mass and lean mass, the mathematical model recommended and validated by the software design team was utilized. Finally, the variables obtained from the analysis were compared with other techniques used in body composition assessment, and associations with the diagnosis of malnutrition by GLIM criteria and diagnosis of sarcopenia by EWGSOP2 criteria were explored.

### 2.3. Statistical Analysis

Continuous variables were presented as mean ± standard deviation. The normality of continuous variables distribution was assessed using the Kolmogorov–Smirnov test. Differences between continuous variables were examined using Student’s *t*-test and, for variables not adhering to a normal distribution, nonparametric tests (Mann–Whitney) were employed. A significance level of *p* < 0.05 was considered for two-tailed tests.

Assessment of the diagnostic performance of variables in detecting malnutrition, sarcopenia, or low muscle mass was conducted using receiver operating characteristic (ROC) curves and calculating the area under the curve (AUC). The accuracy of these measurements was assessed using AUC by plotting sensitivity against 1-specificity. ROC curves were utilized to identify the optimal cut-off values by maximizing the product of sensitivity and specificity. Data analysis was carried out using the SPSS 26.0 program (SPSS Inc., Chicago, IL, USA).

### 2.4. Ethics

All participants provided informed consent prior to their inclusion in the study. The study was conducted in accordance with the Declaration of Helsinki, and the research protocol received approval from the Research Ethics Committee of Malaga on 26 July 2018 (reference number #26072018). The protocol was further validated for the final analysis of CT images and data sharing between centers by the Research Ethics Committee for Medicines of the Vall d’Hebron University Hospital on 29 February 2024 (reference number #PR(AG)489/2021).

## 3. Results

A total of 586 patients were enrolled in the study, with a mean age of 68.4 ± 10.2 years. Among the sample, there was a total of 365 men (62.3%) and 221 women (37.7%). Table 1 provides an overview of the sample’s general characteristics.

### Assessment of Body Composition and Anthropometric Measures

The data from the morphofunctional assessment conducted on the participants, encompassing anthropometry, bioelectrical impedance analysis (BIA), and handgrip strength, are presented in Table 2.

Applying the GLIM criteria for malnutrition diagnosis, it was revealed that 189 patients (32.2%) had experienced a weight loss exceeding 5% in the preceding six months, although only 53 patients (9%) had a low BMI.

Utilizing BIA to assess muscle mass, 55 patients (9.4%) exhibited an FFMI below the specified cut-off points (13.9% of women and 6.7% of men). Consequently, 245 patients (41.8%) were diagnosed with malnutrition according to GLIM criteria (47.5% of women and 38.4% of men).

Applying the EWGSOP2 criteria for sarcopenia diagnosis, 142 patients (24.3%) had a low handgrip strength (27% of women and 22.8% of men). Utilizing BIA, 127 patients (21.6%) exhibited an ASMM below the specified cut-off points (22.2% of women and 21.2% of men). Consequently, 56 patients (9.6%) were diagnosed with sarcopenia (9.7% of women and 9.5% of men).

Patients displayed considerable diversity in body composition as assessed by CT. These findings are depicted in Table 3.

Table 4 presents a comparison of body composition TC parameters based on the patient’s nutritional status and sarcopenia diagnostic.

Using ROC curve analysis for SMI, we found a limited ability to detect malnutrition per GLIM criteria (AUC = 0.60, 95% CI 0.57–0.65, *p* < 0.001). When examining individual phenotypic criteria, low BMI showed good discriminative capacity (AUC = 0.81, 95% CI 0.77–0.86, *p* < 0.001), whereas weight loss did not (AUC = 0.55, 95% CI 0.48–0.61, *p* = 0.192). SMI effectively identified low FFMI (AUC = 0.82, 95% CI 0.77–0.87, *p* < 0.001). The SMI cutoff points for diagnosing low FFMI were 32.75 cm^2^/m^2^ (sensitivity 77%, specificity 64.3%; AUC = 0.79, 95% CI 0.70–0.87, *p* < 0.001) for women and 39.9 cm^2^/m^2^ (sensitivity 77%, specificity 72.7%; AUC = 0.85, 95% CI 0.80–0.90, *p* < 0.001) for men (Figure 2).

Furthermore, we found that SMA had a mild discriminative capacity for detecting sarcopenia based on EWGSOP2 criteria (AUC = 0.71, 95% CI 0.64–0.78, *p* < 0.001). When examining specific criteria, SMA showed limited capacity for detecting low handgrip strength (AUC = 0.68, 95% CI 0.64–0.73, *p* < 0.001) but good capacity for detecting low ASMM (AUC = 0.71, 95% CI 0.65–0.76, *p* < 0.001). The SMA cutoff points for diagnosing low ASMM were 83.2 cm^2^ (sensitivity 76.7%, specificity 55.3%; AUC = 0.77, 95% CI 0.69–0.84, *p* < 0.001) for women and 112.6 cm^2^ (sensitivity 82.3%, specificity 58.6%; AUC = 0.79, 95% CI 0.74–0.85, *p* < 0.001) for men (Figure 3).

## 4. Discussion

Our study has shown that body composition values obtained through CT, particularly SMA and SMI at the L3 level, accurately determine low muscle mass according to both the GLIM criteria for malnutrition and the EWGSOP2 criteria for sarcopenia. We have established gender-specific cutoff points and recommend their adoption for these diagnostic standards.

The prevalence of malnutrition in patients with CRC can be as high as 60% [4]. Previous analysis of our sample from Malaga supported this, showing a malnutrition prevalence of 59.8%, determined by Subjective Global Assessment (SGA) [7]. In our current sample, using GLIM criteria and BIA to determine muscle mass, we found a malnutrition prevalence of 41.8%. This discrepancy could be due to the limited ability of indirect measures like FFMI to detect low muscle mass and the difficulty of BMI in identifying malnourished individuals within an overweight sample (Figure 1). Indeed, a study demonstrated that BMI is less predictive of complications compared with muscle mass [17]. Other studies have shown that structured tools (such as SGA or MNA) are more effective in identifying malnutrition in cases where muscle mass determination is challenging [18]. Given the limited utility of the phenotypic criterion of low BMI in diagnosing malnutrition, the use of new direct muscle assessment techniques such as nutritional ultrasound^®^ [19] or CT, as presented in this study, is justified. Studies comparing CT and BIA have shown that BIA may have systematic errors in determining FFM [20]. CT images for muscle mass assessment are now included in the guidelines for evaluating the muscle mass phenotypic criterion for GLIM diagnosis of malnutrition [21]. Recent studies have followed these recommendations, using CT data, particularly SMI, for implementing GLIM criteria in CRC patients. Consistent with our study, these studies observed lower malnutrition prevalence compared with those using BIA [22]. Furthermore, it has been demonstrated that when applying different tools for the determination of low muscle mass, SMI was the determinant that best correlated with postoperative complications, independent of other phenotypic and etiologic criteria [23].

Low muscle mass is prevalent across various cancer types and disease stages, challenging pre-existing beliefs about the determinants of its prevalence [24]. Previous studies have measured the prevalence of low muscle mass in CRC patients using CT scans. Miyamoto et al. reported a 25% sarcopenia rate in patients with stage I to stage III colorectal cancer, similar to our sample; however, they only considered low skeletal muscle mass as the definition of sarcopenia [14]. Similarly, Souza et al. found a 15% sarcopenia rate in colorectal cancer patients [25]. Previous studies have primarily diagnosed sarcopenia in CRC patients using a combination of SMI and BMI, with muscle strength and body composition rarely evaluated [26].

Few studies have evaluated the prevalence of sarcopenia using the EWGSOP criteria. A study in Brazil examined muscle mass in 31 patients who had previously undergone abdominal CT scans [13]. The prevalence of sarcopenia in this study was 19.4% using the EWGSOP criteria, compared with 48.4% of patients with low SMI on CT, indicating considerably higher values than those observed in our sample. In this case, the analysis of CT images to gather body composition data was conducted using semiautomated software. The determination of low muscle mass was based on the SMI cutoff points defined by Prado et al., which consider low SMI as below 55.4 cm^2^/m^2^ for men and 38.9 cm^2^/m^2^ for women [27]. These cutoff points were established in a study including patients with sarcopenic obesity and solid tumors of the respiratory and gastrointestinal tracts, potentially overestimating the prevalence of low muscle mass. Furthermore, these cutoff points were based on patient survival, whereas sarcopenia is associated with multiple clinical complications [28,29], not solely mortality.

There is a relationship between the loss of quantity and quality of skeletal muscle mass, the systemic inflammatory response, and survival in patients with operable colorectal cancer [30]. Some studies have found that sarcopenia and myosteatosis are negative factors for colorectal cancer patients’ survival [31,32]. Assessing both quantitative and qualitative aspects of muscle mass in body composition evaluation introduces an innovative method in personalized patient therapy. This approach potentially offers a new perspective in colorectal surgery, aiming to reduce postoperative mortality rates, refine surgical planning, and enhance clinical outcomes [33]. In this comprehensive assessment of muscle mass, CT, which includes both quantitative and qualitative aspects, emerges as a tool superior to previous gold standards [34]. However, as shown in Table 4, the HU does not exhibit significant differences between groups with or without sarcopenia, indicating that conventional cutoff points typically do not account for the qualitative aspects of the muscle. Therefore, further studies are needed to understand the relationship between muscle mass determined by CT, both quantitatively and qualitatively, with mortality and other major clinical outcomes.

In our research, we assessed the predictive capacity of muscle mass as determined by CT, both in diagnosing malnutrition and sarcopenia. As anticipated, the unadjusted values of muscle mass do not account for factors such as weight loss, limiting their ability to predict malnutrition. Nevertheless, we contend that the aforementioned cutoff points accurately reflect the concept of low muscle mass. Thus, we propose their utilization in the application of GLIM and EWGSOP2 criteria.

Our study has several strengths. It is a multicenter study with a large number of individuals at similar disease stages, minimizing sample heterogeneity. Additionally, it includes an innovative AI-assisted assessment of body composition, aimed at reducing intra- and interindividual variability. The proposed technique is integrated into routine clinical practice, eliminating the need for additional patient evaluations.

However, our study also has some limitations. As an observational study, the results should be interpreted within the appropriate population context, and causal relationships cannot be established. The recommended cutoff points are based on current consensus measurements derived from techniques like BIA, which measure muscle mass indirectly and may not accurately reflect true muscle mass.

## 5. Conclusions

In conclusion, AI-assisted body composition assessment using CT is proposed as a valuable tool in the morphofunctional evaluation of patients with colorectal cancer prior to surgery. In these patients, who typically undergo prior imaging tests, CT can serve as an accessible tool to provide quantitative and qualitative data on muscle mass for the application of GLIM criteria for malnutrition and the EWGSOP2 sarcopenia criteria, for which we provide cutoff points. By enhancing the evaluation of body composition and nutritional status, these improved morphofunctional assessment tools have the potential to significantly improve clinical outcomes for CRC patients.

## Figures and Tables

**Figure 1 nutrients-16-01869-f001:**
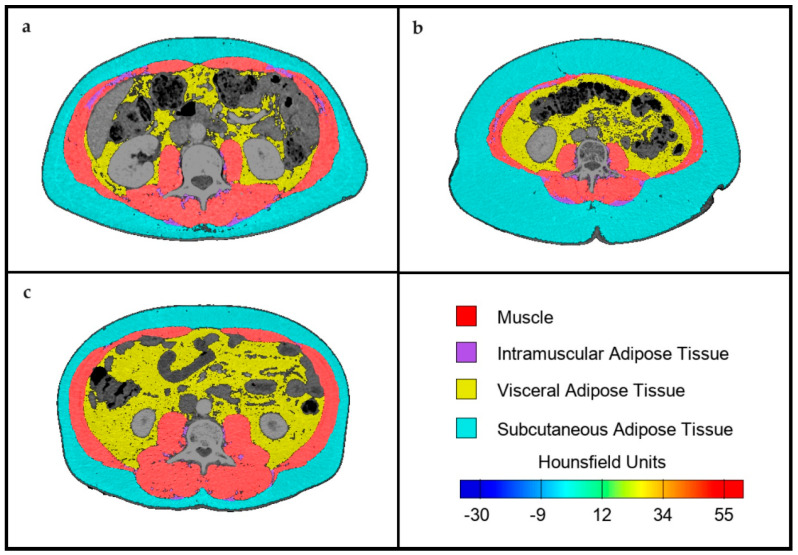
Images extracted from the FocusedON-BC software: (**a**) patient with normal BMI and normal muscle mass; (**b**) patient with elevated BMI and decreased muscle mass; (**c**) patient with elevated BMI and normal muscle mass.

**Figure 2 nutrients-16-01869-f002:**
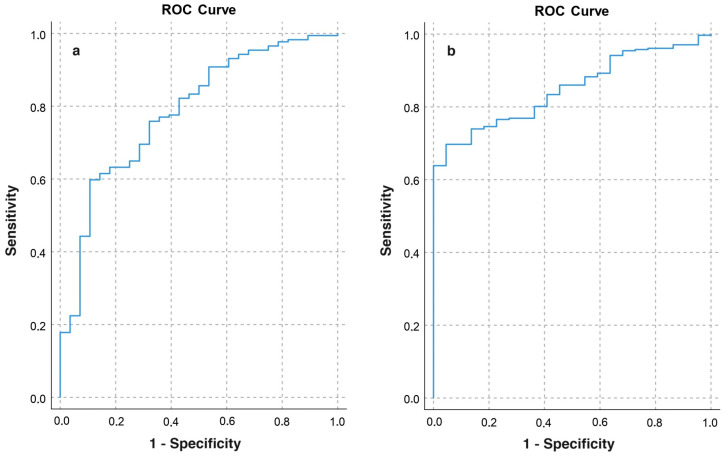
ROC curve analysis. Discriminative capacity of SMI for detecting low FFMI in women (**a**) and men (**b**).

**Figure 3 nutrients-16-01869-f003:**
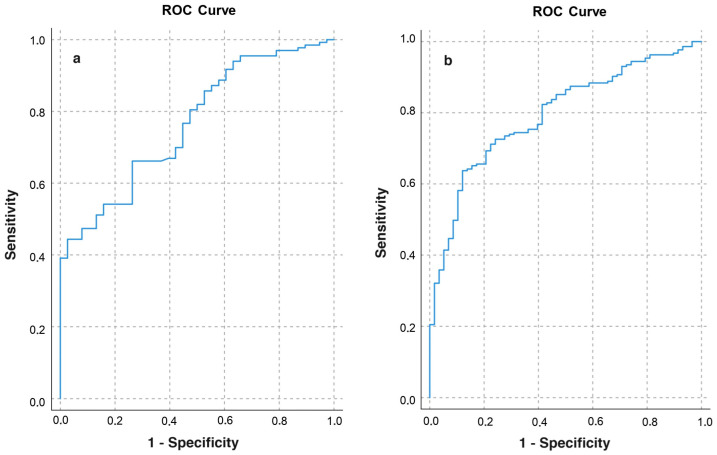
ROC curve analysis. Discriminative capacity of SMA for detecting low ASMM in women (**a**) and men (**b**).

**Table 1 nutrients-16-01869-t001:** General features.

		Total (*n* = 586)	Men (*n* = 365)	Women (*n* = 221)	*p* Value
Age (years)	mean ± SD	68.4 ± 10.2	68.3 ± 10.7	69.1 ± 10.1	0.36
BMI (kg/m^2^)	mean ± SD	27 ± 5.1	27.1 ± 4.7	26.9 ± 5.6	0.63
Type of cancer	*n* (%)				0.93
Colon		423 (72.2)	263 (72.1)	160 (72.4)	
Rectum		163 (27.8)	102(27.9)	61 (27.6)	
Stage	*n* (%)				0.49
Unknown		37 (6.3)	26 (7.1)	11 (5)	
I		102 (17.4)	69 (18.9)	33 (14.9)	
II		184 (31.4)	113 (31)	71 (32.1)	
III		214 (36.5)	126 (34.5)	88 (39.8)	
IV		49 (8.4)	31 (8.5)	18 (8.1)	
Type of surgery	*n* (%)				0.75
Open		84 (14.3)	51 (14)	33 (14.9)	
Laparoscopic		502 (85.7)	314 (86)	188 (85.1)	

Abbreviations: BMI = body mass index; SD = standard deviation.

**Table 2 nutrients-16-01869-t002:** Nutritional status: malnutrition and sarcopenia criteria.

		Total (*n* = 586)	Men (*n* = 365)	Women (*n* = 221)	*p*-Value
BMI (kg/m^2^)	mean ± SD	27.1 ± 5.1	27.1 ± 4.7	26.9 ± 5.9	0.63
Low BMI	*n* (%)	53 (9)	26 (7.1)	27 (12.2)	0.037
Weight loss > 5%	*n* (%)	189 (32.2)	119 (32.5)	70 (31.8)	0.9
FFMI (kg/m^2^)	mean ± SD	19.4 ± 2.9	20.5 ± 2.6	17.6 ± 2.4	<0.001
Low FFMI	*n* (%)	55 (9.4)	24 (6.7)	31 (13.9)	0.006
Malnourished (GLIM criteria)	*n* (%)	245 (41.8)	140 (38.4)	105 (47.5)	0.029
Handgrip strength (kg)	mean ± SD	28.4 ± 10.3	33.6 ± 8.9	19.9 ± 5.9	<0.001
Low handgrip strength	*n* (%)	142 (24.3)	82 (22.8)	60 (27)	0.25
ASMM (kg)	mean ± SD	21.1 ± 4.8	22.9 ± 3.9	16.7 ± 2.6	<0.001
Low ASMM	*n* (%)	127 (21.6)	78 (21.2)	49 (22.2)	0.81
Sarcopenia (EWGSOP2 criteria)	*n* (%)	56 (9.6)	35 (9.5)	21 (9.7)	0.94

Abbreviations: SD: standard deviation; BMI: body mass index; FFMI: fat-free mass Index; ASMM: appendicular skeletal muscle mass.

**Table 3 nutrients-16-01869-t003:** Body composition parameters by TC.

		Men (*n* = 365)	Women (*n* = 221)	*p* Value
Muscle area (SMA) (cm^2^)	mean ± SD	130.96 ± 25.48	90.59 ± 16.39	*p* < 0.001
Muscle percentage	mean ± SD	17.63 ± 4.42	14.36 ± 3.95	*p* < 0.001
Muscle Hounsfield Units	mean ± SD	39.44 ± 9.19	36.37 ± 9.99	*p* < 0.001
SMI (cm^2^/m^2^)	mean ± SD	45.52 ± 8.79	36.79 ± 6.29	*p* < 0.001
IMAT area (cm^2^)	mean ± SD	16.41 ± 11.09	15.81 ± 9.58	*p* = 0.51
IMAT percentage	mean ± SD	2.13 ± 1.47	2.34 ± 1.21	*p* = 0.08
IMAT Hounsfield Units	mean ± SD	−62.89 ± 6.54	−63.06 ± 6.37	*p* = 0.76
VAT area (cm^2^)	mean ± SD	192.57 ± 107.09	180.88 ± 104.19	*p* = 0.19
VAT percentage	mean ± SD	24.19 ± 13.6	26.09 ± 11.71	*p* = 0.09
VAT Hounsfield Units	mean ± SD	−93.79 ± 10.65	−94.48 ± 10.62	*p* = 0.31
SAT area (cm^2^)	mean ± SD	193.58 ± 103.75	177.53 ± 118.22	*p* = 0.09
SAT percentage	mean ± SD	24.59 ± 14.38	24.54 ± 11.89	*p* = 0.96
SAT Hounsfield Units	mean ± SD	−93.79 ± 10.65	−93.68 ± 11.78	*p* = 0.9

SD: standard deviation; SMI: skeletal muscle index, IMAT: intramuscular adipose tissue; VAT: visceral adipose tissue; SAT: subcutaneous adipose tissue.

**Table 4 nutrients-16-01869-t004:** Differences in body composition parameters by TC according to nutritional status and sarcopenia.

		Normo-Nourished (*n* = 341)	Malnourished (*n* = 245)	*p*-Value	Nonsarcopenic (*n* = 514)	Sarcopenic (*n* = 72)	*p*-Value
Muscle area (SMA) (cm^2^)	m ± SD	120.9 ± 29.5	108.6 ± 28.8	*p* < 0.001	115.9 ± 29.8	103.7 ± 23.3	*p* = 0.005
Muscle percentage	m ± SD	15.9 ± 4.2	17.1 ± 4.8	*p* = 0.003	16.2 ± 4.4	18.8 ± 5.8	*p* < 0.001
Muscle HU	m ± SD	38.1 ± 9.1	38.5 ± 10.3	*p* = 0.672	38.5 ± 9.7	39.2 ± 8.4	*p* = 0.592
SMI (cm^2^/m^2^)	m ± SD	43.5 ± 9.2	40.4 ± 8.3	*p* < 0.001	42.3 ± 8.9	36.5 ± 6.4	*p* < 0.001
IMAT area (cm^2^)	m ± SD	17.2 ± 10.8	14.8 ± 10	*p* = 0.006	16.7 ± 11.2	13.5 ± 7.3	*p* = 0.05
IMAT percentage	m ± SD	2.23 ± 1.45	2.16 ± 1.27	*p* = 0.723	2.25 ± 1.48	2.32 ± 1.1	*p* = 0.727
IMAT HU	m ± SD	−63.8 ± 5.9	−61.8 ± 6.9	*p* < 0.001	−63.7 ± 6.6	−61.5 ± 6.2	*p* = 0.028
VAT area (cm^2^)	m ± SD	209.4 ± 104.8	158.6 ± 100.9	*p* < 0.001	193.1 ± 104.2	107.9 ± 71.4	*p* < 0.001
VAT percentage	m ± SD	26.6 ± 13.9	22.6 ± 10.9	*p* < 0.001	25.6 ± 14.1	17.4 ± 9.1	*p* < 0.001
VAT HU	m ± SD	−96.2 ± 7.3	−90.8 ± 11.7	*p* < 0.001	−94.3 ± 9.5	−87.1 ± 11.3	*p* < 0.001
SAT area (cm^2^)	m ± SD	210.9 ± 107	154.9 ± 104.9	*p* < 0.001	189.9 ± 108.9	125.9 ± 82.9	*p* < 0.001
SAT percentage	m ± SD	26.8 ± 14.9	21.4 ± 10.4	*p* < 0.001	24.9 ± 14.8	20.4 ± 10.9	*p* = 0.036
SAT HU	m ± SD	−96.3 ± 9.2	−90.1 ± 12.4	*p* = 0.014	−95.1 ± 10.9	−88.4 ± 14.2	*p* < 0.001

m: mean; SD: standard deviation; HU: Hounsfield units; SMI: skeletal muscle index, IMAT: intramuscular adipose tissue; VAT: visceral adipose tissue; SAT: subcutaneous adipose tissue.

## Data Availability

The original contributions presented in the study are included in the article, further inquiries can be directed to the corresponding author.

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
