# Peer review of "AI-Assisted Body Composition Assessment Using CT Imaging in Colorectal Cancer Patients: Predictive Capacity for Sarcopenia and Malnutrition Diagnosis"

_nutrients, 2024, doi:10.3390/nu16121869_

Round 1

Reviewer 1 Report

Comments and Suggestions for Authors

    • The manuscript clearly states the importance of assessing muscle mass in colorectal cancer (CRC) patients due to its association with complications and prognosis. The dual aim of evaluating AI-assisted L3 CT for body composition and establishing diagnostic cutoff points is well-defined.
    • In the introduction, consider emphasising the potential clinical impact of improved assessment methods on patient outcomes to further highlight the significance.
    • The manuscript provides a comprehensive overview of the study's methodology, including the retrospective design, patient inclusion criteria, and various assessment techniques (anthropometry, BIA, handgrip strength, and AI-assisted CT).
    • Clarify the timeline and geographic scope in a single sentence for better readability. For example: "This retrospective observational study included CRC patients from three regional hospitals in Spain, assessed between October 2018 and July 2023."
    • The results section is detailed, presenting both the prevalence of malnutrition and sarcopenia according to the respective criteria and the performance of CT-derived variables in diagnostic discrimination through ROC analysis.
    • Simplify the statistical details for a more concise presentation. Consider summarizing key findings in a few sentences and using bullet points for specific values if space permits.
    • The conclusion effectively summarizes the utility of AI-assisted CT in the pre-surgical evaluation of CRC patients, highlighting the establishment of specific cutoff points for muscle mass assessment.
    • Reinforce the potential benefits of these findings in clinical practice, such as enhanced diagnostic accuracy and tailored treatment planning, to underscore the relevance of the research.
    • The manuscript is logically structured, with a clear progression from introduction to methods, results, and conclusions. The use of technical terms and criteria (GLIM, EWGSOP2) is appropriate for the target audience.
    • Enhance readability by breaking down long sentences and ensuring each section flows smoothly. For instance, the results could be more digestible with shorter sentences or bullet points for key statistics.

Conclusions: AI-assisted body composition assessment using CT is a valuable tool in the morphofunctional evaluation of CRC patients prior to surgery. CT provides quantitative data on muscle mass for the application of the GLIM and EWGSOP2 criteria, with specific cutoff points established for diagnostic use.

This revised version aims to enhance clarity and readability while maintaining the detailed content and key findings of the original manuscript.

Author Response

Dear Reviewer,

Thank you for your thorough review and constructive feedback on our manuscript. We appreciate your insights and have made the following changes to address your suggestions:

  1. Emphasis on Clinical Impact: In the introduction, we have highlighted the potential clinical impact of improved assessment methods on patient outcomes. We emphasized how these methods can enhance diagnostic accuracy and support tailored treatment plans.

  2. Clarification of Timeline and Geographic Scope: We clarified the timeline and geographic scope in the methodology section, detailing the specific hospitals involved and the assessment period.

  3. Simplification of Statistical Details: We simplified the presentation of statistical details in the results section to improve readability and clarity. This involved summarizing key findings in more concise sentences and ensuring the information is easy to follow.

  4. Reinforcement of Potential Benefits in Clinical Practice: In the conclusion, we reinforced the potential benefits of our findings for clinical practice. We highlighted how our study's results could improve clinical outcomes by enhancing the evaluation of body composition and nutritional status.

  5. Enhancing Readability: We revised the discussion section to improve readability by breaking down long sentences and ensuring smooth transitions between ideas. This included restructuring sentences to make the content more accessible and coherent.

We hope these changes effectively address your suggestions and improve the overall quality and clarity of our manuscript. Thank you again for your valuable feedback.

Best regards,

Reviewer 2 Report

Comments and Suggestions for Authors

The assessment of muscle mass is very important in evaluating the nutritional status of colorectal cancer patients, as decreased muscle mass is linked to higher complications and poorer prognosis. This study evaluates the utility of AI-assisted L3 CT in assessing body composition and determining low muscle mass using the GLIM criteria for malnutrition and the EWGSOP2 criteria for sarcopenia in CRC patients before surgery. The study also aims to establish specific cutoff points for muscle mass in men and women for these diagnostic frameworks. The study, being multicentered, enrolled a good number of 586 patients at similar disease stages, minimizing sample heterogeneity. Also, it includes an innovative AI-assisted assessment of body composition which could be a valuable tool in the future.

I find this study of very good quality.

The limitations that I detect have been also underlined by the authors such as the fact that it is an observational study therefore it cannot establish causal relationships. The proposed cutoff points are based on current consensus measurements, such as BIA, which indirectly measure muscle mass and may not accurately reflect true muscle mass. But as I said all these are mentioned in the limitations section!

Author Response

Dear Reviewer,

Thank you very much for your thorough review and positive feedback on our manuscript. We are pleased to hear that you find our study of good quality and recognize the potential value of our AI-assisted assessment of body composition in colorectal cancer patients.

We appreciate your acknowledgment of the limitations we addressed in the manuscript, such as the observational nature of the study and the basis of our proposed cutoff points on current consensus measurements like BIA.

Your comments reinforce the importance of our work and encourage us to continue our research in this area. Thank you again for your insightful comments and support.

Yours sincerely,

Dr. Francisco José Sánchez Torralvo

Ms. Virginia Soria Utrilla

Dr. Gabriel Olveira

Reviewer 3 Report

Comments and Suggestions for Authors

I read the manuscript carefully. This is undoubtedly an interesting and well-conducted study that evaluated the utility of AI-assisted L3 CT for assessing body composition and determining low muscle mass using GLIM criteria for malnutrition and EWGSOP2 criteria for sarcopenia in CRC patients before surgery. Additionally, the authors aimed to describe cutoff points for muscle mass in men and women and propose their utilization in applying this diagnostic consensus. The strength of this study is that It is a multicenter study with a large number of individuals at similar disease stages, which minimizes sample heterogeneity. In addition, this study includes an innovative AI-assisted assessment of body composition, aimed at reducing intra- and inter-individual variability. Limitations of the study are that is an observational study, the results should be interpreted within the appropriate population context, and causal relationships cannot be established. The recommended cutoff points are based on measurements proposed in the current consensus, derived from techniques like BIA, which measure muscle mass indirectly and may not accurately reflect true muscle mass. To be considered for publication, some minor revisions are required:

Regarding Table 1, the authors could indicate the average age of the two groups, men and women. They should also indicate p-values ​​to express any significant differences between males and females and for the type of cancer, stage, and type of surgery.

Regarding Table 2, the authors could indicate BMI values, weight loss > 5%, and the presence of sarcopenia according to the EWGSOP2 criteria in the two groups, men and women. Also in this case they should indicate the p values ​​to express any significant differences between males and females. Regarding the ROC curves, the authors should insert a figure for each ROC curve or a summary table for all the parameters examined, with the AUC values, confidence intervals, and p-values.

Author Response

Dear Reviewer,

Thank you for your careful review and positive feedback on our manuscript. We appreciate your constructive suggestions and have made the following revisions:

We have added the average age of the two groups, men and women, in Table 1 and included p-values to express any significant differences between males and females, as well as for the type of cancer, stage, and type of surgery. In Table 2, we have included BMI values, weight loss > 5%, and the presence of sarcopenia according to the EWGSOP2 criteria for the two groups, men and women, along with the relevant p-values.

Regarding the ROC curves, we have maintained the most representative figures of our findings and simplified the presentation of the statistical values in the text to ensure clarity and focus on the most relevant results.

We hope these changes address your concerns and enhance the overall quality of our manuscript. Thank you once again for your valuable feedback.

Best regards,
Dr. Francisco José Sánchez Torralvo
Ms. Virginia Soria Utrilla
D. Gabriel Olveira